# ENERGY-EFFICIENT SAMPLING USING STOCHASTIC MAGNETIC TUNNEL JUNCTIONS

## ABSTRACT

(Pseudo)random sampling, a costly yet widely used method in (probabilistic) machine learning and Markov Chain Monte Carlo algorithms, remains unfeasible on a truly large scale due to unmet computational requirements. We introduce an energy-efficient algorithm for uniform Float16 sampling, utilizing a room-temperature stochastic magnetic tunnel junction device to generate truly random floating-point numbers. By avoiding expensive symbolic computation and mapping physical phenomena directly to the statistical properties of the floating-point format and uniform distribution, our approach achieves a higher level of energy efficiency than the state-of-the-art Mersenne-Twister algorithm by a minimum factor of 9721 and an improvement factor of 5649 compared to the more energy-efficient PCG algorithm. Building on this sampling technique and hardware framework, we decompose arbitrary distributions into many non-overlapping approximative uniform distributions along with convolution and prior-likelihood operations, which allows us to sample from any 1D distribution without closed-form solutions. We provide measurements of the potential accumulated approximation errors, demonstrating the effectiveness of our method.

## 1 INTRODUCTION

The widespread implementation of artificial intelligence (AI) incurs significant energy use, financial costs, and $CO_2$ emissions. This not only increases the cost of products, but also presents obstacles in addressing climate change. Traditional AI methods like deep learning lack the ability to quantify uncertainties, which is crucial to address issues such as hallucinations or ensuring safety in critical tasks. Probabilistic machine learning, while providing a theoretical framework for achieving much-needed uncertainty quantification, also suffers from high energy consumption and is unviable on a truly large scale due to insufficient computational resources (Izmailov et al., 2021). At the heart of probabilistic machine learning and Bayesian inference is Markov Chain Monte Carlo (MCMC) sampling (Kass et al., 1998; Murphy, 2012; Hoffman & Gelman, 2014). Although effective in generating samples from complex distributions, MCMC is known for its substantial computational and energy requirements, making it unsuitable for large-scale deployment for applications such as Bayesian neural networks (Izmailov et al., 2021). In general, random number generation is an expensive task that is required in many machine learning algorithms.

To address these challenges, this paper proposes a novel hardware framework aimed at improving energy efficiency, in particular tailored for probabilistic machine learning methods. Our framework builds on uniform floating-point format sampling utilizing stochastically switching magnetic tunnel junction (s-MTJ) devices as a foundation, achieving significant gains in both computational resources and energy consumption compared to current pseudorandom number generators. In contrast to existing generators, this device-focused strategy not only enhances sampling efficiency but also incorporates genuine randomness originating from the thermal noise in our devices. Simultaneously, this noise is crucial for the probabilistic functioning of the s-MTJs and is associated with low energy costs during operation.

We present an acceleration approach for efficiently handling probability distributions. Our experiments confirm its effectiveness by quantifying potential approximation errors. This work does not seek to create a one-size-fits-all setting for all possible probabilistic algorithms that manage proba-

bility distributions. Rather, we offer a solution approach that researchers can customize and utilize based on individual required sampling resolution dependent on a specific algorithm.

Our contributions can summarized as follows:

1. We present a novel, highly energy-efficient stochastically switching magnetic tunnel junction device which is designed to improve both the energy efficiency and precision of our sampling approach. The device is capable of generating samples from a Bernoulli distribution whose parameter $p$ can be controlled using a current bias.

2. We present a closed-form solution that defines the parameters for a collection of Bernoulli distributions applied to the bit positions of the floating-point format, leading to samples that adhere to a distribution without the need for symbolic calculations. Our simulations indicate that this hardware configuration surpasses existing random number generators in terms of energy efficiency by a factor of 5649 when using Float16. Additionally, our method achieves genuine randomness through the use of thermal noise in our hardware devices. In general, this approach is suitable for any entropy source device or even (pseudo)random number generator that can produce bits in a reliable (and efficient) Bernoulli fashion.

3. We propose the representation of arbitrary one-dimensional distributions using a mixture of uniforms model. This approach utilizes our highly efficient hardware-supported uniform sampling approach to enable sampling from arbitrary 1D distributions. We introduce convolution and prior-likelihood transformations for this model to learn and sample from such distributions without closed-form solutions. Our experimental evaluation shows that this method is effective, as evidenced by the small approximation error in KL-divergence when compared to sampling results from known closed-form solutions ($0.0343 \pm 0.1473$ for the convolution and $0.0141 \pm 0.1073$ for prior-likelihood) for basic usage scenarios.

All code of our experiments is available at www.github.com/TBA.

The structure of this paper begins by reviewing relevant work on random number generation and Markov-Chain-Monte-Carlo algorithms for probabilistic machine learning in Section 2. Section 3 provides an introduction to the floating-point format, which is the format utilized for generating samples. In the Approach Section 4, we introduce the stochastically switching magneto-tunneling junction device being utilized in our approach. Following this, we outline a configuration for these devices to generate uniform floating-point samples, addressing the statistical challenge of mapping Bernoulli distributions to specific bitstring positions within the floating-point format. Additionally, we propose our approach for representing, sampling, and converting arbitrary 1D-distributions using a mixture of uniforms as representation. Section 5 illustrates our approach through particular instances and assesses potential approximation errors arising from both the devices and our theoretical framework in the Float16 format. The paper concludes with Section 6, where we summarize our findings and outline further research directions. We used LLM-based tools to improve the writing style and code generation. All reported experiments and simulations can be performed on consumer-grade computers.

## 2 RELATED WORK

A majority of artificial intelligence algorithms rely on random number generators. Random number generators (RNG) are employed for weight initialization or dropout in deep learning or taking random actions in reinforcement learning. In probabilistic machine learning, Markov-Chain-Monte-Carlo (MCMC) algorithms utilize them for sampling from proposal distributions or for making decisions on whether to accept or reject samples based on random draws.

Hence, the research community focused on the development of efficient random number generators (L'Ecuyer, 1994) and their infrastructure (Tan et al., 2021; Nagasaka et al., 2018) shares similarities to this work. Physical (true) random number generators (TRNG) using physical devices is an active research field since the 1950s (L'Ecuyer, 2017). Currently used random number generators are often feasability-motivated free-running oscillators with randomness from electronic noise (Stipcevic & Koç, 2014). A very recent subfield are Quantum Based Random Number generators (QRNG) (Mannalatha et al., 2023; Józwiak et al., 2024; Stipcevic & Koç, 2014; Herrero-Collantes & Garcia-Escartin, 2017). The concept of employing stochastic magnetic tunnel junctions for random

number generation has been investigated in recent years. Although these methods generally outperform traditional algorithmic random number generators in terms of energy efficiency, they lack three crucial features for machine learning applications that our approach addresses. First, they lack the ability to directly produce results using the floating-point format (Zhang et al., 2024; Chen et al., 2022; Oosawa et al., 2015; Perach et al., 2019), which is critical for machine learning applications. Converting results to floating-point format later (Fu et al., 2023) introduces unnecessary overhead, reducing energy efficiency. In general, the unequal spacing characteristic of the floating-point format complicates the transition from integers, making it non-trivial to maintain all possible floating-point number candidates within a specific distribution. Second, most works lack the flexibility to generate arbitrary distributions. Zhang et al. (2024) propose using a conditional probability table for this purpose. However, their method involves adjusting the current bias for each bit in a sample and repeating this process for every required sample, which substantially increases energy consumption. In addition, sequential operations that scale with the number of bits reduce the achievable sampling speed. Furthermore, they address integer generation only, making their work unsuitable for machine learning applications. Finally, none of the works addresses directly sampling from a product of likelihoods (distributions) as often encountered in probabilistic machine learning. It should be noted that our conceptual approach can in principle be applied with any RNG that generates parametrizable Bernoulli distributions, given that they are sufficiently (energy-)efficient.

MCMC methods like Metropolis-Hastings (Hastings, 1970; Metropolis et al., 1953) and the state-of-the-art Hamiltonian Monte Carlo (HMC) (Neal et al., 2011) algorithm are crucial for this research. The use of MCMC for Bayesian inference and probabilistic machine learning represents the core application area of this paper, aiming to achieve computational and energy-efficient deployment at a large scale. Furthermore, (pseudo)random number generation is often discussed in the context of Monte Carlo approaches as they are closely intertwined and take advantage of efficient random number sampling as proposed by us. On the other hand, we propose an alternative hardware-supported approach to the MCMC algorithms themselves with our mixture model. In general, our approach differs from traditional pseudorandom number generation of MCMC algorithms as we employ a genuinely random sampling method, making it less suitable for scenarios requiring reproducibility (L'Ecuyer, 2017; Holohan, 2023; Hill, 2015) or reversability (Yoginath & Perumalla, 2018), our objectives align in efficient random number generation and genuine statistical independence.

Antunes & Hill (2024) accurately measured the energy usage of random number generators (Mersenne-Twister, PCG, and Philox) in programming languages and frameworks such as Python, C, Numpy, Tensorflow, and PyTorch, thus providing a quantification of energy consumption in tools relevant to AI. The energy measurements of this benchmark serve as baseline against our approach.

## 3 PRELIMINARIES

We use the floating-point format as the number representation of interest as this is also the format that machine learning algorithms use. We define a generic floating-point number as follows:

$$x = \pm 2^{e-b} \cdot d_1.d_2 \ldots d_t, \tag{1}$$

where $e$ is the exponent adjusted by a bias $b$, $d_1.d_2 \ldots d_t$ represent the mantissa, $d_i \in \{0, 1\}$, and $d_1 = 1$ indicates an implicit leading bit for normalized numbers.

While our approach is generally applicable to any floating-point format, we demonstrate the approach for the Float16 format in this paper. The use of the Float16 format compared to formats with more precision bits is advantageous in a real-world setting as it demands less rigor in setting the current bias for the s-MTJ devices, which is especially relevant for higher-order exponent bits.

In the following, we describe a Float16 number by its 16-bit organization

$$B = (b_0, b_1, \ldots, b_{15}), \tag{2}$$

where $b_{15}$ is the sign bit, $b_{14}$ to $b_{10}$ are the exponent bits with a bias of 15, and $b_9$ to $b_0$ are the mantissa bits. The implicit bit remains unexpressed. This arrangement represents the actual storage format of the bits in memory. By expressing the floating-point format in terms of its bit structure, we can directly map an s-MTJ device's output bit to its equivalent position in the Float16 format.

## 4 APPROACH

### 4.1 PROBABILISTIC SPINTRONIC DEVICES

Spintronic devices are a class of computing (logic and memory) devices that harness the spin of electrons (in addition to their charge) for computation (Žutić et al., 2004). This contrasts with traditional electronic devices which only use electron charges for computation. In essence, we interpret the upwards and downwards electronic spin as binary states instead of their charge. Changing state corresponds to changing the direction of the spin. The field of spintronics holds potential for lowering energy consumption in comparison to conventional electronics. Applying insufficient current results in the electronic spin states exhibiting probabilistic behavior due to ambient temperature. In this research, we utilize this probabilistic behavior by aligning it directly with algorithmic requirements.

Spintronic devices are built using magnetic materials, as the magnetization (magnetic moment per unit volume) of a magnet is a macroscopic manifestation of its correlated electron spins. The prototypical spintronic device, called the magnetic tunnel junction (MTJ), is a three-layer device which can act both as a memory unit and a switch (Moodera et al., 1995). It consists of two ferromagnetic layers separated by a thin, insulating non-magnetic layer. When the magnetization of the two ferromagnetic layers is aligned parallel to each other, the MTJ exhibits a low resistance ($R_P$). Conversely, when the two magnetizations are aligned anti-parallel, the MTJ exhibits a high resistance ($R_{AP}$). By virtue of the two discrete resistance states, an MTJ can act as a memory bit as well as a switch. In practice, the MTJs are constructed such that one of the ferromagnetic layers stays fixed, while the other layer's magnetization can be easily toggled (free layer, FL). Thus, by toggling the FL, using a magnetic field or electric currents, the MTJ can be switched between its '0' and '1' state.

An MTJ can serve as a natural source of randomness upon aggressive scaling, i.e. when the FL of the MTJ is shrunk to such a small volume that it toggles randomly just due to thermal energy in the vicinity. It is worth noting that the s-MTJ can produce a Bernoulli distribution like probability density function (PDF), with $p = 0.5$, without any external stimulus, by virtue of only the ambient temperature. However, applying a bias current across the s-MTJ can allow tuning of the PDF through the spin transfer torque mechanism. As shown in Figure 5c-f of Appendix A, applying a positive bias current across the device makes the high resistance state more favorable, while applying a negative current has the opposite effect. In fact, by applying an appropriate bias current across the s-MTJ, using a simple current-mode digital to analog converter as shown in Figure 6a of Appendix A, we can achieve precise control over the Bernoulli parameter ($p$) exhibited by the s-MTJ. The $p$-value of the s-MTJ responds to the bias current through a sigmoidal dependence. A more detailed version of this section on the physical principles, device structure and simulations of the s-MTJ device can be found in Appendix A.

### 4.2 RANDOM NUMBER SAMPLING

This section describes the configuration of s-MTJ devices representing Bernoulli distributions for generating uniform random numbers in floating-point formats, particularly Float16. To apply this method to other floating-point formats, modify the number of total bits in Equation 3, 5 and 6 as well as the number of exponent bits in Equation 8 and their positions in the format in variable $e$ of Equation 6.

The configuration $C$ for a set of s-MTJ devices is defined as follows:

$$C = \{(b_i, p_i) \mid p_i \in [0, 1], b_i \in \{b_0, \ldots, b_{15}\}\}, \tag{3}$$

where each $p_i$ is the parameter of a Bernoulli distribution representing the probability of the corresponding Float16 format bit being '1' in the output.

The goal is to configure $C$ so that, with infinite resampling, the sequence $B_n$ of Float16 values converges to a uniform distribution $D$ over the full format. Formally, we seek $C$ such that:

$$\lim_{n \to \infty} P(B_n = b \mid C) = D(b), \text{ where } D = \text{Uniform}(-65504, 65504) \tag{4}$$

In order to meet this condition, we need to assign each bit position $b_i$ of the Float16 format a probability $p_i$, representing the frequency of each bit's occurrence in a uniform Float16 distribution

Table 1: Required 1-bit occurrences in a 3-bit exponent representation

| | **1-Bit Count** | | | | | | | |
|---|---|---|---|---|---|---|---|---|
| $e_3$ | 0 | 0 | 0 | 0 | $2^4$ | $2^5$ | $2^6$ | $2^7$ |
| $e_2$ | 0 | 0 | $2^2$ | $2^3$ | 0 | 0 | $2^6$ | $2^7$ |
| $e_1$ | 0 | $2^1$ | 0 | $2^3$ | 0 | $2^5$ | 0 | $2^7$ |

(Equations 5-8). The mantissa bits are assigned a value of 0.5, as detailed in line 6, ensuring uniformity across the range they cover. This method extends to the sign bit, whose equal likelihood of toggling maintains the format's symmetry.

In floating-point formats, increasing the exponent doubles the range covered by the mantissa due to the base 2 system. Higher exponent ranges need more frequent sampling to maintain uniform coverage, as simply doubling sample occurrence from one range to the next does not preserve uniformity. Table 2 shows the number of 1-bits for each exponent in a 3-bit example. In general, one can see a specific overall pattern. Specifically, $e_1$ has four groups of size 1, $e_2$ has two groups of size 2, and $e_3$ has one group of size 1. More generally, the first count of any exponent group is always $2^{2^{i-1}}$. For the first exponent, groups are size 1 (excludable by $\mathbf{1}_{\{i>1\}}$). For other exponents, remaining 1-Bit counts in the first group are $\sum_{k=1}^{c-1} 2^{2^{i-1}+k}$, where $c = 2^{i-1}$ is the group size, depending on the position $i$ in the floating-point format. The count of groups based on bit position $i$ and total bits $e$ is $z = 2^{-i+e}$. The count sums for remaining groups are given by $\sum_{k=1}^{z-1}\sum_{g=1}^{c-1} 2^{2^{i-1}+2^i \cdot k+g}$, where $z$ is the number of groups and $c$ their size. The highest exponent bit $e_3$ with one group is excluded using $\mathbf{1}_{\{z>1\}}$. To find the probability of 1-Bit occurrences for each exponent $e_i$, divide by the total bits $2^{(2^e)} - 1$, which depends on the exponent bits $e$.

Combining everything, we derive the equation for the configuration $C$ as follows:

$$C = \{(b_i, p_i) \mid p_i \in [0,1], b_i \in \{b_0, \ldots, b_{15}\}, \text{ where} \tag{5}$$

$$p_i = \begin{cases} \frac{o_i - 9}{2^{(2^e)}-1} & \text{if } i \in \{10, \ldots, 14\}, \\ 0.5 & \text{otherwise} \end{cases}, \text{ and} \tag{6}$$

$$o_i = 2^{2^{i-1}} + \sum_{k=1}^{c-1} 2^{2^{i-1}+k} \cdot \mathbf{1}_{\{i>1\}} + \sum_{k=1}^{z-1} 2^{2^{i-1}+2^i \cdot k} + \sum_{k=1}^{z-1}\sum_{g=1}^{c-1} 2^{2^{i-1}+2^i \cdot k+g} \cdot \mathbf{1}_{\{z>1\}}, \text{ and} \tag{7}$$

$$z = 2^{-i+e}, c = 2^{i-1}, e = 5. \tag{8}$$

After obtaining a sample $s$, min-max normalization can be applied to linearly transform it into a sample $s'$ that adheres to any specified uniform distribution within the Float16 range:

$$s' \sim \text{Uniform}(a,b) = a + \frac{(s+65504) \cdot (b-a)}{131008}. \tag{9}$$

The transformation must be performed in a format exceeding Float16, like Float32 or a specialized circuit, to maintain numerical stability and precision, due to exceeding Float16 limits in the denominator of Equation 9. We assume special cases like NaNs differently represented and Infinities discarded; we do not evaluate convention specifics in this paper.

### 4.3 SAMPLING AND LEARNING ARBITRARY 1D-DISTRIBUTIONS

This section addresses how to represent and sample from any arbitrary one-dimensional distribution, aiming for random and energy-efficient non-parametric sampling without closed-form solutions.

Sampling from a uniform distribution within the Float16 range is an energy-efficient method. Given that hardware representations of continuous distributions are inherently discretized during real computations, we use a mixture model of uniform distributions as distributional representation. This approach (Gao et al., 2022) is well-established for handling real-world data that standard distributions do not adequately represent. In general, mixture models of all forms are used in probabilistic machine learning to approximate multimodal and complex distributions (Murphy, 2012). We break down a distribution into several non-overlapping uniform distributions, where the approximation

error depends on the interval size. The weights of these components indicate the relative probability density of each interval within the overall distribution.

Let $D$ be the distribution to be represented, $\mathbf{F}_{16}$ the set of Float16 values, and $U_i \sim \text{Uniform}(a_i, b_i)$ for $i = 1, 2, \ldots, k$ non-overlapping interval components of our mixture model, where each $U_i$ is uniform on $[a_i, b_i)$ with $a_i, b_i \in \mathbf{F}_{16}$. The mixture probability density function $f_U$ is defined by

$$D(x) = f_U(x) = \sum_{i=1}^{k} w_i f_{U_i}(x) \tag{10}$$

such that $\sum_{x \in X} w_i f_{U_i}(x) = 1$ and $w_i f_{U_i}(x)$ is the probability density function of component $U_i$:

$$f_{U_i}(x) = \begin{cases} \frac{1}{b_i - a_i} & \text{if } x \in [a_i, b_i) \\ 0 & \text{otherwise.} \end{cases}$$

To draw a sample from the distribution $D$, we first perform a uniform sampling within the interval $[0, 1]$, which is assigned to the intervals of the components according to their respective weights. From the selected component interval, we then perform another uniform sampling within that specific range. Therefore, obtaining a sample from $D$ requires two uniform sampling steps.

Our approach is particularly suited for the concentration of statistical distributions in ranges (e.g., near zero due to data normalization). Using a high component resolution in this range ensures precise sampling, though it may cause inaccuracies further afield. We propose using a balanced number of s-MTJ devices to manage errors, offering a viable and energy-efficient solution. More research is needed to tailor distribution resolutions to specific algorithms. The effectiveness of our method is demonstrated through the analysis of cumulative approximation errors in Section 5.3.

Probabilistic machine learning relies heavily on thorough sampling from the posterior distribution. We have introduced an efficient sampling method, but operations involving two arbitrary distributions are necessary to derive a posterior distribution. Modern probabilistic machine learning mainly uses distributions that have closed-form solutions and methods for approximating unknown distributions to familiar ones. We introduce both the sum (convolution) and the computation of prior-likelihood (pointwise multiplication) as methods to facilitate the learning of posterior distributions in a non-parametric manner, bypassing the need for closed-form solutions.

In all definitions, it is assumed that the intervals $\{[a_i, b_i]\}$ in our mixture models are consistent across all represented distributions. Variations in notation (e.g., $\{[c_i, d_i]\}$) highlight different distributions.

The convolution $Z = X + Y$ for two independent random variables $X$ and $Y$ is defined as $f_Z(z) = \int_{-\infty}^{\infty} f_X(x) f_Y(z - x) \, dx$. For approximating the convolution using interval sets with weights, we calculate the mean of sums of interval bounds for each combination (Cartesian product). Let $\{X_i = ([a_i, b_i), w_i)\}_{i=1}^{n}$ and $\{Y_j = ([c_j, d_j), v_j)\}_{j=1}^{n}$ represent the mixture models for $X$ and $Y$ respectively, covering the entire Float16 range.

Calculating the means results in

$$m_{ij} = \frac{a_i + b_i}{2} + \frac{c_j + d_j}{2}, \tag{11}$$

with a combined weight:

$$u_{ij} = w_i \cdot v_j. \tag{12}$$

This intermediate set $\{(m_{ij}, u_{ij})\}_{i,j=1}^{n}$ contains pairs of mean and weight. Define $\{Z_l = ([g_l, h_l), r_l)\}_{l=1}^{n}$ as the desired distribution. Update the weights for $Z_l$ by

$$r_l = \sum_{l=1}^{n} u_{ij} \cdot \mathbf{1}_{[g_l, h_l)}(m_{ij}), \tag{13}$$

where $\mathbf{1}_{[g_l, h_l)}(x)$ is the indicator function that is 1 if $x \in [g_l, h_l)$ and 0 otherwise. Lastly, the weights are normalized

$$r_l' = \frac{r_l}{\sum_{s=1}^{n} r_s}. \tag{14}$$

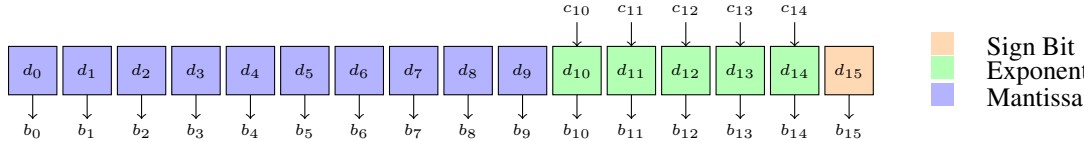

Figure 1: Hardware setup for sampling one value from a uniform Float16 distribution.

It should be noted that sampling from both normalized and unnormalized distributions yields equivalent results since normalization maintains the relative proportions within our distributions. However, it keeps the weights bounded and manageable for storage purposes.

Intermediate pairs of mean and weight for the prior-likelihood computation is obtained by

$$m_{ii} = \frac{a_i + b_i}{2} = \frac{c_i + d_i}{2}, \text{ where } a_i = c_i \text{ and } b_i = d_i \tag{15}$$

and

$$u_{ii} = w_i \cdot v_i. \tag{16}$$

Aside from above equations, the remaining algorithm is that of convolution. Note that the joint distribution is derived by simultaneously sampling from two mixture models. The components of the models remain unchanged during sampling.

## 5 EVALUATION

### 5.1 ENERGY CONSUMPTION OF THE s-MTJ APPROACH

Figure 1 depicts our hardware configuration for sampling a single Float16 value. Each $d_i$ is an s-MTJ device. The devices $d_{10}, \cdots, d_{14}$ for the exponent are equipped with 4 control bits to adjust the current bias $c_i$, which corresponds to the Bernoulli probability. The other devices are set to a fixed current bias equivalent to a Bernoulli of $0.5$. The resolution, which determines how accurately we can set the Bernoulli distributions for a device, is dependent on the number of control bits and is visualized in Figure 2. This Figure displays the specific Bernoulli values achievable with 4 control bits. Although additional control bits could allow for more precise settings, we restrict this number to 4 due to physical limitations in setting current biases in hardware with higher resolution while keeping the bias circuit simple (and hence energy-efficient). Our approach focuses on achieving high accuracy around a probability of 1 (cf. configuration in Section 5.2) by taking advantage of the characteristics of the sigmoid function, thus making 4 bits sufficient for achieving the required probability density function.

For our specific case, where the s-MTJs are being configured to generate a uniform distribution of Float16 samples, the $p$ for each s-MTJ is predetermined and fixed. All the mantissa and sign bits require $p = 0.5$, which is exhibited by the s-MTJ without any current bias (cf. 4.1 and 4.2). Thus, these eleven s-MTJs do not require a current biasing circuit. The predetermined $p$-values for the five exponent bits correspond to specific current biases as shown in Figure 2, which amount to a total power consumption of $20.86\,\mathrm{W}$, as determined through SPICE simulations (see Appendix D). For a sampling rate of $1\,\mathrm{MHz}$, this corresponds to $20.86\,\mathrm{pJ}$ biasing energy per Float16 sample. Reading the state of all sixteen s-MTJs, assuming a nominal resistance of $1\,\mathrm{k\Omega}$ and $10\,\mathrm{ns}$ readout with $10\,\mathrm{\mu A}$ probe current, amounts to an additional readout energy dissipation of $16\,\mathrm{fJ}$ per Float16 sample.

Given a hardware accelerator-style architecture, our system is designed with an embarrassingly parallel structure, capable of producing samples every $1\,\mathrm{\mu s}$. Energy-wise, there is no difference between parallel and sequential setups. Using min-max normalization, sampled intervals can be transformed efficiently into other intervals. It is reasonable that each of the five floating point operations mentioned in Equation 9 within a normalization circuit consumes about $150\,\mathrm{fJ}$ on modern microprocessors (Ho et al., 2023), leading to an extra energy cost of $750\,\mathrm{fJ}$ per sample.

Consequently, generating $2^{30}$ samples without linear transformation yields an energy consumption

$$(16 \cdot 1\,\mathrm{fJ} + 20.862\,\mathrm{pJ}) \cdot 2^{30} = 22.42\,\mathrm{mJ}. \tag{17}$$

Applying the transformation yields

$$(16 \cdot 1\,\text{fJ} + 20.862\,\text{pJ} + 750\,\text{fJ}) \cdot 2^{30} = 23.22\,\text{mJ}. \tag{18}$$

Our method's energy usage is compared to actual energy measurements taken by Antunes & Hill (2024). They benchmarked advanced pseudorandom number generators like Mersenne Twister, PCG, and Philox. This includes evaluations across original C versions (O2 and O3 suffixes refer to C flags) and adaptations in Python, NumPy, TensorFlow, and PyTorch, relevant platforms and languages for AI. Each measurement reports the total energy used to produce $2^{30}$ pseudorandom 32-bit integers or 64-bit doubles, which are common outputs from these generators. Often, specific algorithms and implementations are limited to producing only certain numeric formats (like integers or doubles), particular bit sizes, or specific stochastic properties. As such, comparing different implementations and floating-point formats is somewhat limited. However, given that all implementations serve the same machine learning algorithms and that our energy consumption estimates show vast differences, this comparison is deemed both reasonable and significant.

Although our method introduces considerable energy costs due to transformations, the overall energy usage, when including linear transformations, is reduced by factor 5649 (pcg32integer) compared to the most efficient pseudorandom number generator currently available. Compared to the double-generating Mersenne-Twister (mt19937arO2), we obtain an improvement by factor 9721. We provide a full comparison against all benchmarked generators in Figure 7 of Appendix E.

Quantifying the energy-saving potential impact on downstream tasks is challenging due to the vast array of algorithmic sampling approaches, corresponding domain-specific applications, and possible assumptions at the circuit or software implementation levels. Therefore, we illustrate the potential by comparing the fundamental MCMC rejection sampling approach with our mixture-based sampling method described in Section 4.3. In this benchmark, we focus only on the fundamental operations performed by each algorithm and corresponding energy expenses, making as few assumptions as possible. We ignore related factors such as memory usage, bus transfers, or other implementation specifics. Rejection sampling is a popular MCMC method that enables flexible sampling from arbitrary distributions in an algorithmic manner without additional assumptions, making it an equivalent alternative to our approach. We show the utilized pseudocode in Figure 1 and the resulting energy benchmarks in Figure 8 and 9 of Appendix F. As target distribution, we used a the prior-likelihood product of a $\text{Beta}(2, 5)$ and a $\mathcal{N}(0.1, 0.1^2)$ (cf. Figure 16 of Appendix H). We repeated experiments 100 times with 50 000 samples each and report mean values. We assign both the s-MTJ approach and the rejection sampling approach the same energy costs of 150 fJ for floating-point operations (Ho et al., 2023), including the probability density function for simplification. Our approach utilizes two random uniform draws per sample and according linear transformations. Rejection sampling utilizes two random draws per iteration to decide on a candidate sample: one draw from a proposal distribution (uniform in our experiments) and one uniform draw to determine whether to accept the candidate. It also adds several floating-point operations per iteration. We quantify the potential energy associated with uniform drawings in two ways. First, we consider the rejection sampling algorithm using the `mt19937arO3` random number generator (Mersenne Twister), which is the most energy-efficient floating-point generator in our reference benchmark (see Figure 7 in Appendix E). Second, we assume that the rejection sampling algorithm employs our efficient uniform sampling approach. The benchmark illustrates that sampling from a non-parametric distribution using our method not only offers energy savings but also provides algorithmic improvements. Notably, while rejection sampling does not yield a sample in every iteration, our mixture-based approach consistently does. Overall, Figure 8 in Appendix F shows that the most energy-intensive operation in rejection sampling is the generation of uniform random draws. Comparing the traditional rejection sampling implementation against our s-MTJ approach yields an overall improvement by several orders of magnitude (improvement factor of $5.67 \times 10^{13}$). Even when the rejection sampling algorithm utilizes our s-MTJ approach for uniform draws, we still experience a significant overhead (improvement factor 5.32) due to the inherent inefficiency of rejection sampling. Naturally, more proposals are rejected than accepted (by factor of 5.4x in our experiments), increasing both the necessary random draws and the corresponding arithmetic operations. This demonstrates not only a significant energy-efficiency improvement but also highlights the algorithmic advantage of our mixture-based s-MTJ approach.

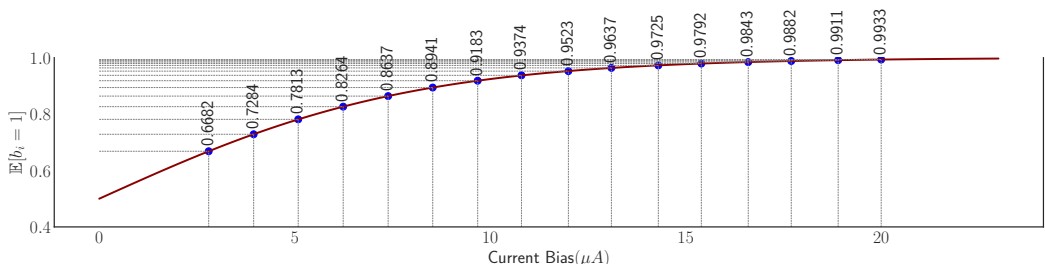

Figure 2: Possible Bernoulli resolutions for s-MTJ device with 4 control bits.

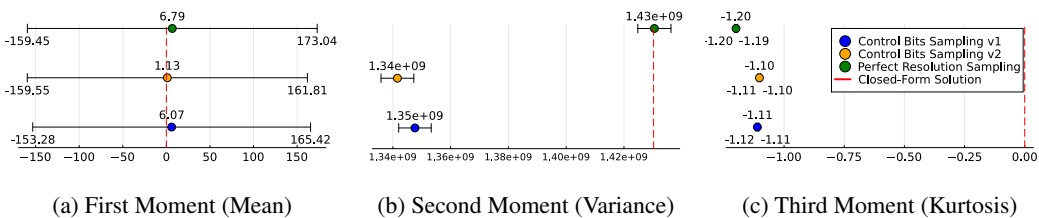

Figure 3: Physical approximation error comparison for the first three moments of the uniform distribution (s-MTJ-based approach vs. closed-form solution sampling). Second moment standard deviation omitted due to equivalence to the means.

## 5.2 PHYSICAL APPROXIMATION ERROR: IMPACT OF CONTROL BITS RESOLUTION

The number of control bits in an s-MTJ device impacts both energy consumption and the precision of setting the energy bias, which in turn affects the available probabilities of obtaining bit samples. Figure 2 illustrates this relationship. This section evaluates the approximation error caused by imprecision in achieving a desired Bernoulli distribution.

Four control bits allow 16 distinct, uniformly spaced current biases for an s-MTJ device. The stability of reading a '1' or '0' from the device follows a sigmoid function, enhancing resolution near 0 and 1, but reducing it around 0.5. This effect is beneficial as it yields the configurations $c_{10}, c_{11}, \cdots, c_{14} = \{(10, 0.66666\overline{6}), (11, 0.80000), (12, 0.94118), (13, 0.99611), (14, 0.99998)\}$ for our hardware setup shown in Figure 1, as derived from Equations 5-8. Higher exponent bits demand greater precision than lower ones, highlighting the advantages of the Float16 format over larger formats due to the physical constraints in setting the energy bias. To precisely analyze distribution shifts, we compared the first three moments (mean, variance, kurtosis) of the uniform Float16 distribution in Figures 3a, 3b, and 3c. We conducted $100\,000$ samples per measurement, repeating each measurement 100 times, and report the results as mean and standard deviations. We evaluated the empirical moments of these distributions against theoretical expectations using closed-form solutions. Control Bits Sampling v1 uses the closest distance, assigning equal probabilities of 0.9933 to $c_{13}$ and $c_{14}$. Control Bits Sampling v2 assigns probabilities of 0.9911 to $c_{13}$ and 0.9933 to $c_{14}$, testing whether having a difference is more effective than the closest distance method (see Figure 2). The mean values over all three moments are consistent for all bit resolutions. Furthermore, the deviation in the second moment is relatively minor given its high absolute value in the closed-form expression. Figure 11-14 of Appendix G visualizes samples using perfect resolution sampling and sampling that considers physical control bit boundaries. The distributions with approximation offsets show a slight bias, favoring values near zero (this is experimentally attributable to the offsets in exponent 4 and 5). However, this primarily accounts for only two bins in the overall range, each representing 0.25% of values. While the overall distribution remains unaffected, the effect can be removed by rejecting samples from the two bins in question, impacting approximately every 200th sample. These observations highlight that physical inaccuracies have minor effects. If necessary, these can be easily addressed through rejection from those bins, depending on the application's requirements. Although we assume that most applications will not be significantly affected, performance evaluations are required to verify this assumption (for any minor distribution shifts).

Table 2: Approximation error comparison (mixture-based approach vs. closed-form solution)

| Approach for Distribution P | $D_{\mathrm{KL}}(P \parallel Q_{\text{closed-form}})$ | $\Delta\, Q_{\text{closed-form}}$ |
|---|---|---|
| Sampling $Q_{\text{closed-form}}$ Convolution | $1.6932 \pm 0.1032$ | - |
| Mixture-based Convolution | $1.7274 \pm 0.1033$ | $0.0343 \pm 0.1473$ |
| Sampling $Q_{\text{closed-form}}$ Prior-Likelihood | $0.7959 \pm 0.0859$ | - |
| Mixture-based Prior-Likelihood | $0.8099 \pm 0.0845$ | $0.0141 \pm 0.1073$ |

### 5.3 Conceptual Approximation Error: Impact of Mixture Model Components

This section discusses approximation errors induced by our conceptual approach due to interval resolution and transformation errors. It examines the convolution and prior-likelihood transformation of two distributions. The convolution analysis spans the interval $[-1, 1)$ with a $0.0005$ resolution, comprising $4000$ elements. Similarly, the prior-likelihood transformation is analyzed over the interval $[-0.5, 1.5)$ using the same resolution.

The approximation error is quantified by setting up transformations as described in Section 4.3. Control bit errors are not considered, attributing the error solely to the theoretical approach. We set up input distributions and their closed-form probability density functions. We convolved two Gaussian distributions $\mathcal{N}(0.2, 0.1^2)$ to get $\mathcal{N}(0.4, 0.1^2 + 0.1^2)$. Using a Beta$(2, 5)$ prior and a $\mathcal{N}(0.1, 0.1^2)$ likelihood, we derived the final distribution by multiplying their densities.

We evaluated the difference in outcomes between our mixture-based approach and the closed-form solution using Kullback-Leibler (KL) divergence. We used kernel density estimation with a uniform kernel and a bandwidth of $0.0005$ for density estimation. To assess the inherent offset between closed-form densities and sampling-based ones due to limited sample sizes, we sampled $50\,000$ times from the Gaussian closed-form distribution. We also used rejection sampling with a uniform proposal distribution, allowing us to obtain samples from the prior-likelihood multiplication. Remaining KL discrepancies can be attributed to the approximation errors of our mixture model. We repeated these sampling-based evaluations $100$ times, recording the mean and standard deviation.

Table 2 shows the approximation errors observed. As shown in Table 2, each method aligns well with the closed-form probability densities. The approximation errors due to sample size are $0.0141 \pm 0.1073$ for prior-likelihood transformations and $0.0343 \pm 0.1473$ for convolutions. The slightly higher error in convolutions is likely due to more frequent recalculations of means and weights (Cartesian product), while prior-likelihood transformations are linear (pointwise multiplication). Appendix 15 illustrates the sampled distributions for these calculations.

## 6 Conclusion and Future Work

We introduced a hardware-driven highly energy-efficient acceleration method for transforming and sampling one-dimensional probability distributions, using stochastically switching magnetic tunnel junctions. This method includes a precise initialization for these devices for uniform random number sampling that beats current state-of-the-art Mersenne-Twister by a factor of $5649$, a uniform mixture model for distribution sampling, and convolution and prior-likelihood computations to enhance learning and sampling efficiency.

We assessed the approximation error associated with the s-MTJ devices and our theoretical framework. Findings show that the physical approximation error is negligible when sampling uniform random numbers. Furthermore, the KL-divergence showed only minor variations compared to sampling from the closed-form solution, noting deviations of $0.0343 \pm 0.1473$ in convolution and $0.0141 \pm 0.1073$ in prior-likelihood operations. Our approach improves existing machine learning algorithms directly by generating random numbers with high efficiency. It also allows the development of specialized solutions designed for specific algorithms and tasks in the future. Further studies will explore the performance impact of approximation and conceptual error on specific algorithms in (probabilistic) machine learning currently unsuitable for MCMC methods and validate the s-MTJ method by building a prototype including statistical randomness testing of the device (Martínez et al., 2018; L'Ecuyer, 2017).

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

## A    ADDITIONAL INFORMATION ON THE SPINTRONIC DEVICE

Spintronic devices are a class of computing (logic and memory) devices that harness the spin of electrons (in addition to their charge) for computation. This contrasts with traditional electronic devices which only use electron charges for computation. Spintronic devices are built using magnetic materials, as the magnetization (magnetic moment per unit volume) of a magnet is a macroscopic manifestation of its correlated electron spins. The prototypical spintronic device, called the magnetic tunnel junction (MTJ), is a three-layer device which can act both as a memory unit and a switch (Žutić et al., 2004; Moodera et al., 1995). It consists of two ferromagnetic layers separated by a thin, insulating non-magnetic layer. When the magnetization of the two ferromagnetic layers is aligned parallel to each other, the MTJ exhibits a low resistance ($R_P$). Conversely, when the two magnetizations are aligned anti-parallel, the MTJ exhibits a high resistance ($R_{AP}$). By virtue of the two discrete resistance states, an MTJ can act as a memory bit as well as a switch. In practice, the MTJs are constructed such that one of the ferromagnetic layers stays fixed, while the other layer's magnetization can be easily toggled (free layer, FL). Thus, by toggling the FL, using a magnetic field or electric currents, the MTJ can be switched between its '0' and '1' state.

An MTJ can serve as a natural source of randomness upon aggressive scaling, i.e. when the FL of the MTJ is shrunk to such a small volume that it toggles randomly just due to thermal energy in the vicinity. As schematically illustrated in Figure 4a, the self-energy of the magnetic layer is minimum and equal for the magnetization pointing vertically up or down, i.e. polar angle $\theta_M = 0^o$ or $180^o$, respectively. The self-energy is maximum for the horizontal orientation ($\theta_M = 90^o$). The corresponding energy barrier, $\Delta E$ dictates the time scale at which the magnet can toggle between the up and down oriented states owing to thermal energy. This time scale follows an Arrhenius law dependence (Camsari et al., 2019), i.e.

$$\tau_{\uparrow\downarrow} = \tau_0 e^{\frac{\Delta E}{kT}}, \tag{19}$$

where, $\tau_0$ is the inverse of attempt frequency, typically of the order of 1 ns, $k$ is the Boltzmann constant and $T$ is the ambient temperature. The energy barrier for a magnet is $\Delta E = K_U V = \mu_0 H_K M_S V/2$, where $K_U$, $V$, $H_K$ and $M_S$ are the magnet's uniaxial anisotropy energy, volume, effective magnetic anisotropy field and saturation magnetization, respectively. $\mu_0$ is the magnetic permeability of free space. Thus, it can be observed that by reducing the volume $V$ of the magnetic free layer, we can make its $\Delta E$ comparable to $kT$ and achieve natural toggling frequencies of computational relevance, as shown in Figure 4b. Figure 5a shows a time-domain plot of the normalized state of such an s-MTJ, calculated using micromagnetic simulations with the MuMax3 package (Vansteenkiste et al., 2014). Further details on the micromagnetic simulations are included in Appendix B. A histogram of the resistance state of this s-MTJ is presented in Figure 5b. It is worth noting that the s-MTJ can produce such a Bernoulli distribution like probability density function (PDF), with $p = 0.5$, without any external stimulus, by virtue of only the ambient temperature. However, applying a bias current across the s-MTJ can allow tuning of the PDF through the spin transfer torque mechanism (Stiles & Zangwill, 2002). As shown in Figure 5c-f, applying a positive bias current across the device makes the high resistance state more favorable, while applying a negative current has the opposite effect. In fact, by applying an appropriate bias current across the s-MTJ, using a simple current-mode digital to analog converter as shown in Figure 6a, we can achieve precise control over the Bernoulli parameter ($p$) exhibited by the s-MTJ. Details on the current-biasing circuit are included in Appendix D. The $p$-value of the s-MTJ responds to the bias current through a sigmoidal dependence, as shown in Figure 6b.

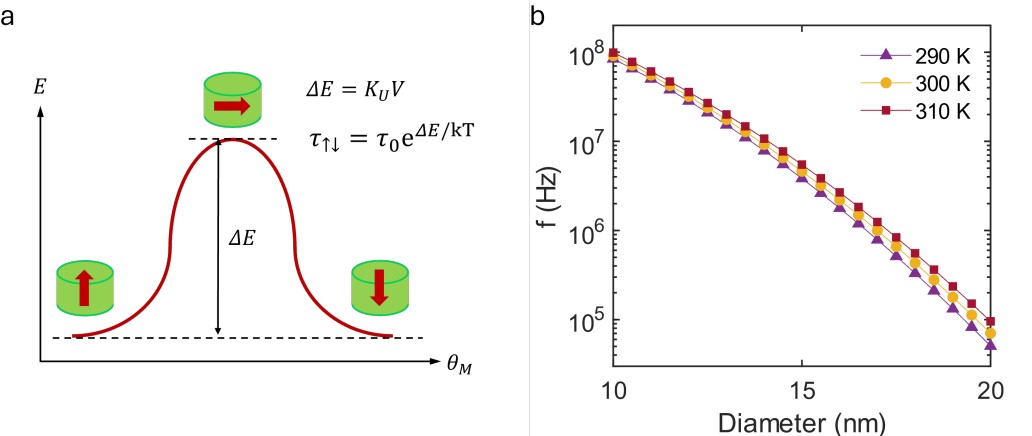

Figure 4: (a) Schematic illustration of the self-energy ($E$) of a nanomagnet with respect to the polar angle ($\theta_M$) of its magnetization (indicated by thick arrows). (b) Natural frequency of stochastic switching for a nanomagnet of a particular diameter at different temperatures.

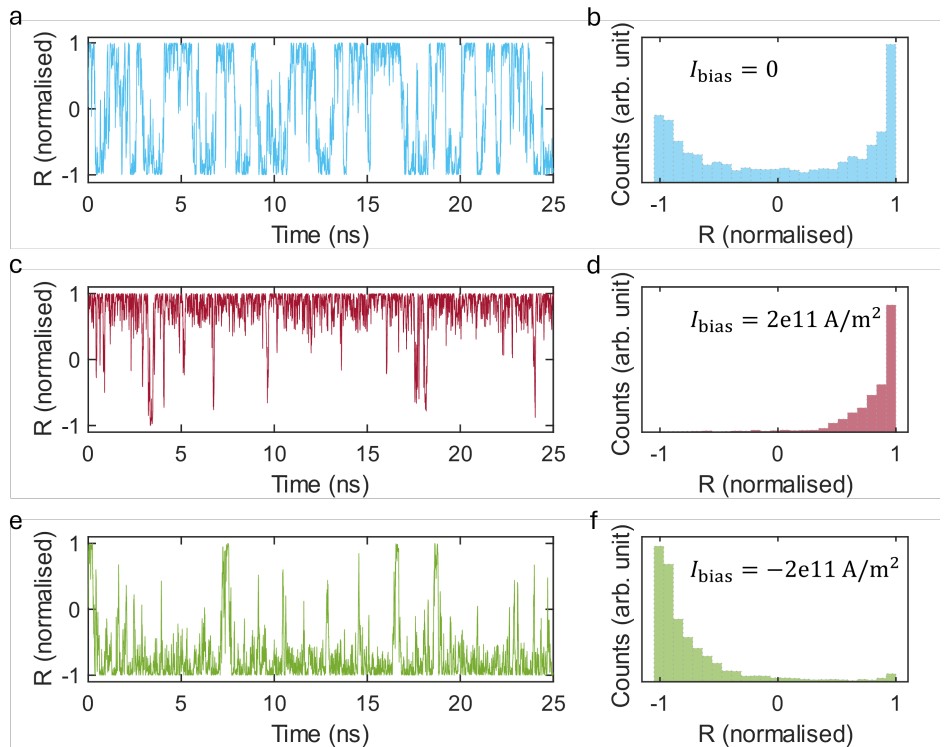

Figure 5: Dynamics of the normalized resistance of a stochastic MTJ for different bias current densities. (a) $I_{\text{bias}} = 0$ produces equal probability of observing the high or low state. (b) Histogram of the observed resistance state for $I_{\text{bias}} = 0$. (c, d) Trace and histogram of the observed resistance for a bias current of $2 \times 10^{11}$ A/m$^2$. (e, f) Trace and histogram of the observed resistance for a bias current of $-2 \times 10^{11}$ A/m$^2$.

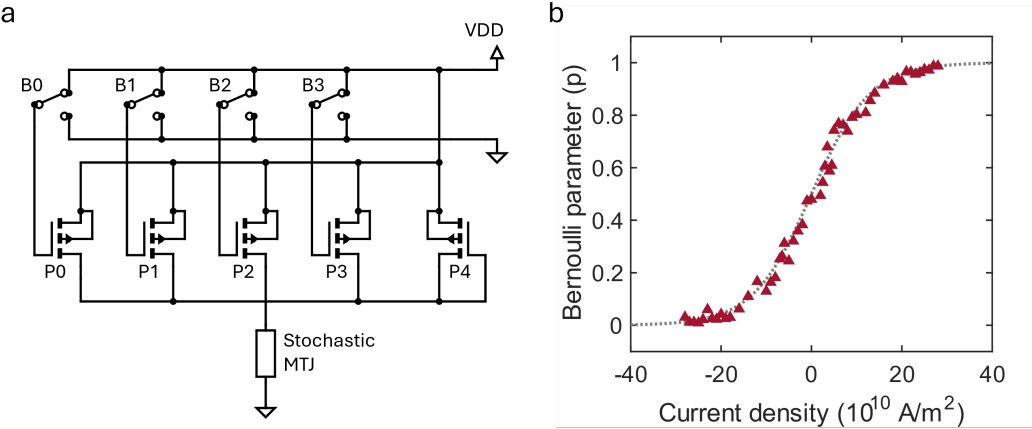

Figure 6: (a) Schematic diagram of a current-mode digital to analog converter for providing the biasing current to a stochastic MTJ. (b) Variation of the Bernoulli parameter of the stochastic MTJ with bias current. Red triangles are data point obtained from micromagnetic simulations, while the grey dotted line is a theoretical fit (sigmoid function).

## B  MICROMAGNETIC SIMULATIONS

Dynamics of a ferromagnet's magnetization in response to external stimuli, like magnetic fields, currents or heat can be modelled using micromagnetic simulations. The magnetization dynamics can be described using a differential equation, known as the Landau-Lifshitz-Gilbert-Slonczewski (LLGS) equation:

$$\frac{d\vec{m}}{dt} = -\gamma \vec{m} \times \vec{H}_{\text{eff}} + \alpha \vec{m} \times \frac{d\vec{m}}{dt} + \tau_{\parallel} \frac{\vec{m} \times (\vec{x} \times \vec{m})}{|\vec{x} \times \vec{m}|} + \tau_{\perp} \frac{\vec{x} \times \vec{m}}{|\vec{x} \times \vec{m}|} \tag{20}$$

where, $\vec{m}$ is the normalized magnetization ($\vec{M}/|\vec{M}|$), $\gamma$ and $\alpha$ are the gyromagnetic ratio and damping constant for the ferromagnet, $x$ is a unit vector along the direction of applied electric current and, $\tau_{\parallel}$ and $\tau_{\perp}$ are current-induced torque magnitudes acting parallel and perpendicular to the current. $\vec{H}_{\text{eff}}$ is the effective magnetic field acting on the ferromagnet, which contains contributions from externally applied magnetic fields, exchange interactions, magneto-crystalline anisotropy, shape anisotropy, thermal fields, and demagnetization, among others.

The simulations results presented here are performed for a van der Waals (vdW) magnetic material, $Fe_3GaTe_2$ (FGaT) (Zhang et al., 2022; Kajale et al., 2024). Being a vdW material, FGaT has a layered structure which makes it an ideal candidate for building ultra-thin (monolayer) magnetic thin films of high quality needed for achieving stochasticity. FGaT also exhibits perpendicular magnetic anisotropy, which means its self-energy is lower for magnetization pointing out of plane as compared to the magnetization pointing in-plane. This property is crucial for building compact, nanoscale spintronic devices. The simulations are performed using the MuMax3 program (Vansteenkiste et al., 2014), for devices shaped as circular discs. The values of different physical parameters used in the micromagnetic simulations are compiled in Table 3. Certain parameters, whose experimental values are not determined, are set to typical values for similar materials and are indicated as such. All simulations can be replicated using standard consumer-grade computers without requiring extensive resources.

Table 3: Parameters Used in Micromagnetic Simulations With the MuMax3 Code.

| Parameter | Value |
|---|---|
| Saturation magnetization ($M_S$) | $3.95 \times 10^4$ A/m (Kajale et al., 2024) |
| Effective anisotropy field ($K_U$) | $3.02 \times 10^6$ A/m (Kajale et al., 2024) |
| Permeability of free space ($\mu_0$) | $1.26 \times 10^{-6}$ kg·m/s$^2$·A$^2$ |
| Temperature ($T$) | 300 K |
| Gilbert damping constant ($\alpha$) | 0.02 (typical) |
| Exchange stiffness ($A_{\text{ex}}$) | $1.3 \times 10^{13}$ J/m |
| Thickness | 1 nm |
| Diameter | 2 nm |

## C    POTENTIAL LIMITATIONS OF SIMULATED s-MTJ DEVICES

While the proposed s-MTJ devices show great promise for energy-efficient true random number generation, their practical implementation remains an active area of research. The materials system integral to achieving reliable device performance at extreme scaling—particularly 2D magnetic materials—presents unique challenges due to their relative novelty. Key hurdles include the wafer-scale growth of room-temperature monolayer 2D magnetic materials with BEOL compatibility and their integration with tunnel barriers (e.g., 2D hBN or bulk MgO) and spin-orbit torque layers. These challenges remain unmet at a wafer scale. Nonetheless, the promising benchmarking results presented in this study may serve as a catalyst for experimental advancements toward realizing these hardware goals. Additionally, we must consider the effects of process-voltage-temperature (PVT) variations on s-MTJs. Leading semiconductor foundries have already established mature MTJ fabrication processes for embedded MRAM (e.g., high-level caches), demonstrating the feasibility of fabricating PVT-robust MTJ devices for commercial applications. However, for s-MTJs specifically, temperature variations may have unique implications. Unlike traditional deterministic MTJs, the natural frequency of stochastic switching in s-MTJs is highly temperature-dependent. To ensure uncorrelated samples, the sampling frequency must remain below the device's natural frequency across the entire rated operating temperature range. It is worth noting, however, that under typical operating conditions, devices are likely to experience heating, which increases the natural frequency of the devices. This inherent behavior provides a safety margin, ensuring that the samples remain uncorrelated even in elevated temperature conditions.

## D  POWER ESTIMATION OF THE CURRENT BIASING CIRCUIT

The current biasing circuit was simulated using Cadence Virtuoso using the Global Foundries 22FDX (22 nm FDSOI) process design kit. The circuit has been designed for a maximum bias current of 20 $\mu$A to attain an s-MTJ with Bernoulli parameter $p = 0.99$. The current levels corresponding to $p = 0.67$ and $p = 0.99$ are divided into 4-bit resolution (Figure 2). The four bias bits (B0-B3) are fed to the transistors P0, P1, P2, P3 (LSB to MSB), which are sized to produce currents $I_0$, $2I_0$, $4I_0$ and $8I_0$, respectively, when the corresponding bias bit it '1'. A constant current $I_{base} = 2.82$ $\mu$A is additionally supplied through P4 to create a baseline of $p = 0.67$ for the s-MTJs. The transistors are operated at a low supply voltage of 0.35 V to achieve a small $I_0 = 1.14$ $\mu$A. Thus, each exponent bit can be set to its requisite Bernoulli parameter by appropriately setting the 4-bit bias word, and the power dissipation in the biasing circuit can be estimated for each of the exponent bits. Lengths of all the transistors are set to 20 nm. Width of P4 is set to 260 nm, while the widths of P0, P1, P2 and P3 are 100 nm, 200 nm, 400 nm, and 800 nm, respectively. As discussed in the main text, our proposed method requires only positive current biases for the stochastic MTJs. Thus, the unipolar current mode DAC proposed here suffices for our application. For more general use cases where both positive and negative bias currents may be needed, a bipolar current-steering DAC can be utilized.

# E ENERGY CONSUMPTION OF RANDOM NUMBER GENERATORS

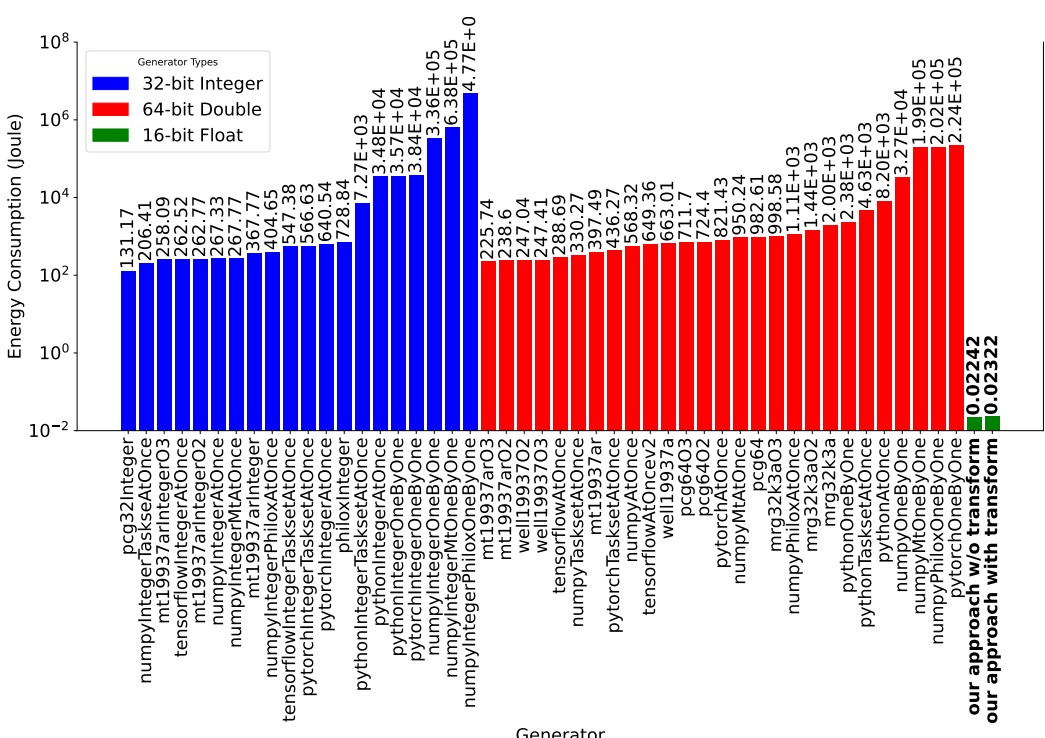

Figure 7: Power consumption analysis in Joules (logarithmic scale) for $2^{30}$ random numbers. Benchmarks were performed by Antunes & Hill (2024).

# F ENERGY CONSUMPTION OF REJECTION SAMPLING

---

**Algorithm 1** Rejection Sampling Algorithm (cf. Koller (2009))

---

**Require:** Probability density $\text{pdf}(\cdot)$ of target distribution, constant $c$, number of samples $N$
**Ensure:** Array of samples $S$ with size $N$ from the target distribution
1: Initialize empty list of samples: $S \leftarrow [\,]$
2: **while** length of $S < N$ **do**
3:     $x_{\text{proposed}} \sim U(0, 1)$
4:     $p_{\text{accept}} \leftarrow \dfrac{\text{pdf}(x_{\text{proposed}})}{c}$
5:     $u \sim U(0, 1)$
6:     **if** $u < p_{\text{accept}}$ **then**
7:         $S \leftarrow S \cup x_{\text{proposed}}$
8:     **end if**
9: **end while**
10: **return** $S$

---

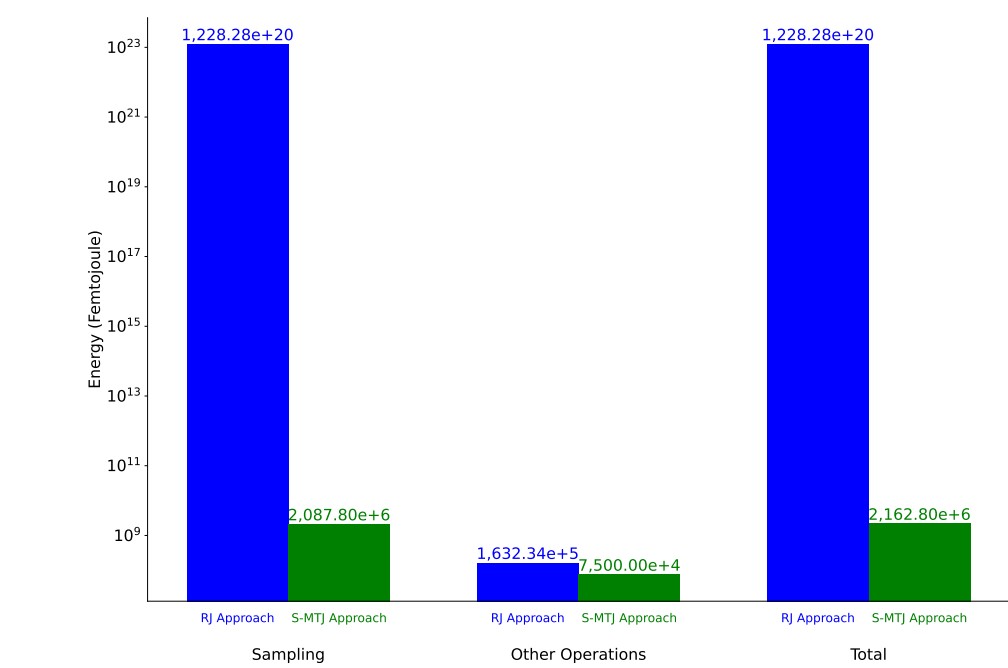

Figure 8: Back-of-the-envelope power consumption analysis in femtojoules (logarithmic scale) for 50 000 samples from rejection sampling (RJ) and the mixture-based sampling approach. RJ sampling assumes draws using mt19937ar03 according to benchmarks from Antunes & Hill (2024). Other operations of the S-MTJ approach refer to the normalization overhead.

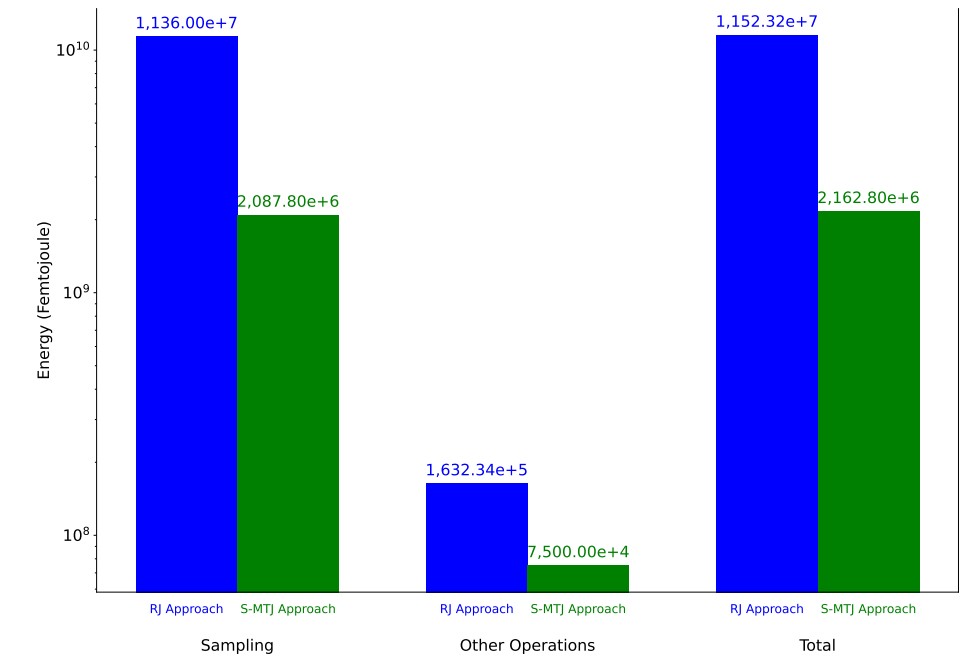

Figure 9: Back-of-the-envelope power consumption analysis in femtojoules (logarithmic scale) for 50 000 samples from rejection sampling (RJ) and the mixture-based sampling approach. RJ sampling assumes draws using s-MTJs. Other operations of the S-MTJ approach refer to the normalization overhead.

# G  ADDITIONAL FIGURES ON PHYSICAL APPROXIMATION ERROR

Figure 10: Visualization of samples obtained with three different assumptions. Perfect Resolution Sampling assumes the precise values obtained from Equations 5-8 in Section 4.2. Control Bits Sampling v1 assumes the closest distance measure to actual obtainable control bits. Control Bits Sampling v2 assumes that each exponent bit should actually be different over closest distance, even if the physically closest distance would imply redundant values.

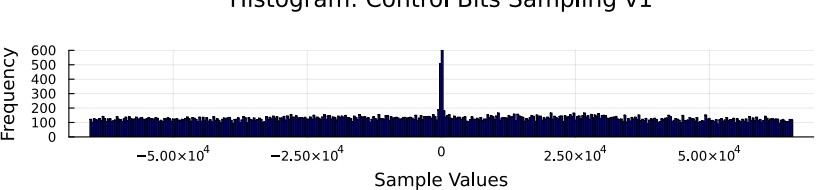

Figure 11: Histogram of $100\,000$ samples with $400$ bins over the full Float16 range obtained by Perfect Resolution Sampling.

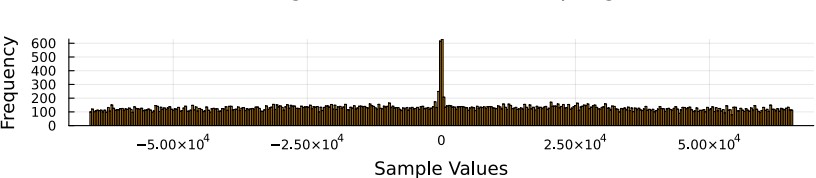

Figure 12: Histogram of $100\,000$ samples with $400$ bins spanning the full Float16 range obtained via Control Bits Sampling v1. The values show a slight bias, favoring those near zero. Each bin represents 0.25% of the overall range. Flattening the distribution by rejecting samples from the two most overrepresented bins would affect only 0.5 % of samples.

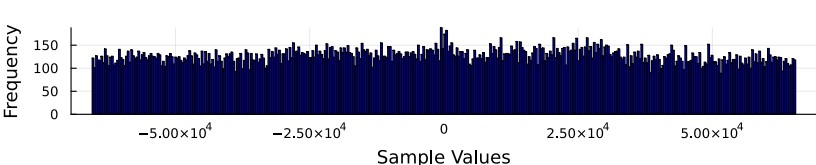

Figure 13: Histogram of $100\,000$ samples with $400$ bins spanning the full Float16 range obtained via Control Bits Sampling v2. The values show a slight bias, favoring those near zero. Each bin represents 0.25% of the overall range. Flattening the distribution by rejecting samples from the two most overrepresented bins would affect only 0.5% of samples.

Figure 14: Histogram of $100\,000$ samples with $400$ bins spanning the full Float16 range obtained via Control Bits Sampling v1 with rejecting from the two most overrepresented bins around zero.

## H    ADDITIONAL FIGURES FOR CONCEPTUAL APPROXIMATION ERROR

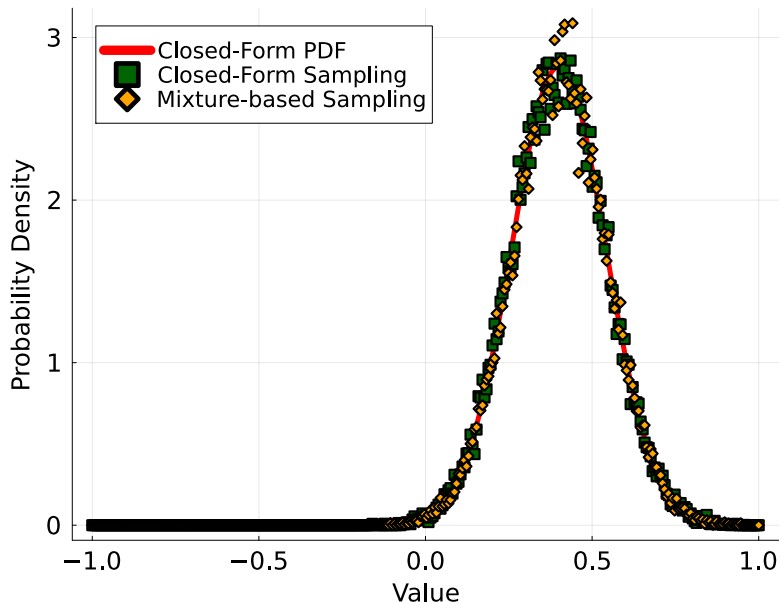

Figure 15: Sampling from the convolution of two Gaussian distributions, $\mathcal{N}(0.2, 0.1^2)$ and $\mathcal{N}(0.2, 0.1^2)$, resulting in $\mathcal{N}(0.4, \sqrt{0.1^2 + 0.1^2})$.

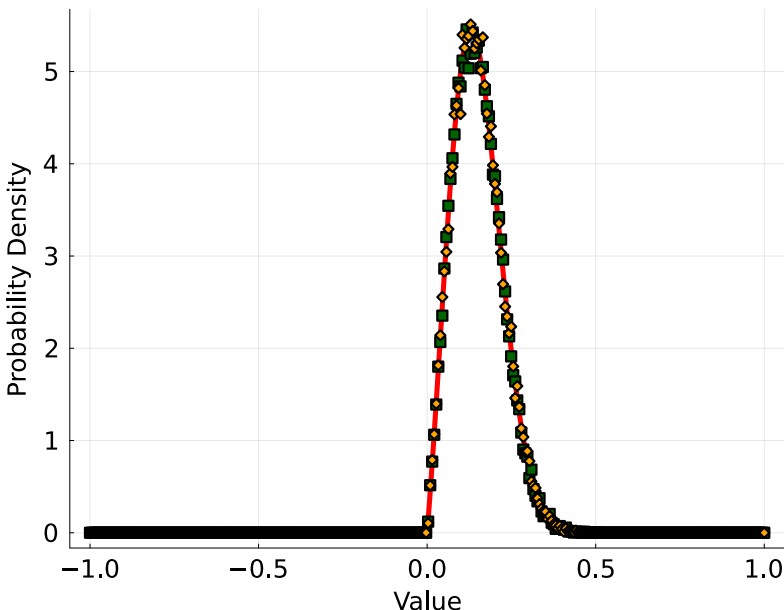

Figure 16: Sampling after Prior-Likelihood Transformation: Using a $\text{Beta}(2, 5)$ prior and a $\mathcal{N}(0.1, 0.1^2)$ likelihood, the final distribution is derived by multiplying their densities.

