# OpenReview forum: "Energy-Efficient Sampling Using Stochastic Magnetic Tunnel Junctions"
_ICLR.cc/2025/Conference — Submitted to ICLR 2025_

### Official Review · Reviewer_bvc4 · 2024-10-28

**Soundness:** 2
**Presentation:** 3
**Contribution:** 3
**Rating:** 8
**Confidence:** 3

**Summary:**

The authors propose a hardware framework based on spintronic devices. By aggressively scaling the device down to nanoscale, thermal noise dominates the resistance state to enable randomness. In addition, the randomness can be biased by applying current through the device to accommodate arbitrary probability p in Bernoulli distribution. The authors use the above features to facilitate 16-bit floating point uniform sampling.

**Strengths:**

Sampling is essential in probabilistic models. The idea of using physical devices for efficient sampling is promising.

**Weaknesses:**

The devices do not seem to be readily deployable and may face manufacturing issues, thus may require further development.

**Questions:**

1. Sampling from a uniform distribution is the primary objective of this work and I greatly appreciate the authors' efforts in evaluating the error. Still, would it be possible to illustrate the actual distribution that the system is sampling from, given the limited number of control bits and the precision requirements indicated in Equation 6? So readers can have intuitive understandings. It would be great to show some zoom-in subfigures for detailed pdf variation in less-precise areas, too.
2. For a 32-bit floating point, how would the actual distribution be like? How many control bits would it require to achieve sufficient sampling precision for 32-bit float?
3. Since the probability of each bit can be atomically manipulated, is it feasible to use the system to sample an arbitrary distribution for arbitrary data format? I find this potential very intriguing.

---

> ### Author Response · Authors · 2024-11-22
> **Response**
>
> Dear Reviewer bvc4,
>
> Thank you very much for your review and for your positive assessment!
>
> *The devices do not seem to be readily deployable and may face manufacturing issues, thus may require further development.*
>
> We agree that the hardware devices are not readily deployable as this is a prototypical stage. **However, we think this is not a weakness of our paper, but to be expected conducting research on very novel devices and this principle applies to any approach that utilizes dramatically novel hardware devices.** To emphasize that eventual deployment is a realistic pathway, we can point to deterministic MTJs, which, with similar materials and device structures, have already been adopted into commercial computer chips in the form of MRAMs.
> It is important to note that the prototypical manufacturing of these devices takes several years and requires significant material resources, especially in an academic setting. Considering these factors, we believe that sharing and comparing our results is both justifiable and beneficial for the broader community, particularly given the notable differences we have observed.
>
> *Sampling from a uniform distribution is the primary objective of this work and I greatly appreciate the authors' efforts in evaluating the error. Still, would it be possible to illustrate the actual distribution that the system is sampling from, given the limited number of control bits and the precision requirements indicated in Equation 6? So readers can have intuitive understandings. It would be great to show some zoom-in subfigures for detailed pdf variation in less-precise areas, too.*
>
> Our paper includes such a straightforward visualization in Figure 10 in Appendix F. However, the sampled and ideal distributions are so similar that no differences can be seen visually. In that regard, Figure 10 effectively conveys the intuition that there is no significant approximation error. To provide more precision in quantifying the error while maintaining the underlying intuition, we included Figures 3a-c in the main body of the paper, which visualize the first three moments of the uniform distribution and respective errors.
>
> *Since the probability of each bit can be atomically manipulated, is it feasible to use the system to sample an arbitrary distribution for arbitrary data format? I find this potential very intriguing.*
>
> It is possible to sample any arbitrary distribution in floating-point format. While converting the floating-point format to other formats after sampling is possible, it is not part of the scope of this paper.
>
> Kind regards,
>
> The Authors

---

> > ### Comment · Reviewer_bvc4 · 2024-11-23
> >
> > Thank you for providing the figure in Appendix F. Here are a few further comments.
> >
> > 1. I agree with Reviewer LDN7, that a limitation section would be beneficial.
> > 2. In Figure 10, could you please show the sampled distribution in histogram with sufficient number of bars? Which may appear to be a more standard way to show distribution.

---

> > > ### Author Response · Authors · 2024-11-26
> > > **Feedback Incorporation**
> > >
> > > Dear Reviewer bvc4,
> > >
> > > Thank you for your feedback, which we have incorporated.
> > >
> > > *In Figure 10, could you please show the sampled distribution in histogram with sufficient number of bars? Which may appear to be a more standard way to show distribution.*
> > >
> > > Histograms for all presented variants have been added in Appendix G. This also revealed two overrepresented ranges due to control bits approximation, which we addressed in Section 5.2.
> > >
> > > *I agree with Reviewer LDN7, that a limitation section would be beneficial.*
> > >
> > > We added a limitations section. Please see Appendix C.
> > >
> > > Kind regards,
> > >
> > > The Authors

---

### Official Review · Reviewer_LDN7 · 2024-10-30

**Soundness:** 3
**Presentation:** 3
**Contribution:** 3
**Rating:** 8
**Confidence:** 4

**Summary:**

Authors present a device capable of sampling one-dimensional distributions with arbitrary distributions, in float16 precisions. They describe how their device is more efficient at this task than standard hardware, and provide benchmarks which indicate orders of magnitude of energy efficiency gains through simulations of their hardware, both through cadence and through custom micromagnetic simulations.

**Strengths:**

The paper is well written, and the approach is clearly explained.
The idea is scientifically sound, and the benchmarks are extensive and include very recent work (2024) that compares samplers. Two types of simulations were considered: i) through cadence, with the global foundries PDK, which includes realistic effects (though it is unclear what is/isn't included), and ii) through custom micromagnetic simulations, which are shown to match well theoretical predictions. I am not expert in these devices but it seems to me they have shown significant promise for a few years and this paper pushes the idea further and seems like a solid contribution to the field of sampling.

**Weaknesses:**

One weakness is that the device was only simulated and not actually realized, so uncertainty remains with respect to practical performance   (although the simulations are extensive, so I do not think this is a reason to reject).
I think the paper would benefit from a Limitations section, which would make it clear what effects were not captured in the simulations. For example, what about PVT variations?

Another weakness that may be improved on is the energy efficiency gain on downstream tasks, i.e., you show dramatic improvement over digital samplers, but how much impact would this have on a downstream task, say of a MCMC sampling task of a given probability distribution? This would be interesting to run with a software with pyro and show the overall energy efficiency improvement.

**Questions:**

Did you think about an extension to multidimensional probability distributions? I understand that this is a much more difficult problem, but it would be interesting to at least discuss it in the paper. Maybe some form of correlations may be implementable between s-MTJs which would allow for sampling of a given class of multidimensional probability distributions.

---

> ### Author Response · Authors · 2024-11-21
> **Updated Paper with Downstream Benchmark & Elaboration on Simulation Uncertainty**
>
> Dear Reviewer LDN7,
>
> thank you very much for your time reviewing the paper and your feedback.
>
> *Another weakness that may be improved on is the energy efficiency gain on downstream tasks, i.e., you show dramatic improvement over digital samplers, but how much impact would this have on a downstream task, say of a MCMC sampling task of a given probability distribution? This would be interesting to run with a software with pyro and show the overall energy efficiency improvement.*
>
> We completely agree that the energy efficiency gains in downstream tasks are highly interesting. We have implemented your suggestion to compare a MCMC sampling task for a specific probability distribution against our approach. **We have updated the PDF version of the paper, and you can find the benchmark results described in lines 400-431, which are supported by Appendix Section E (including pseudocode and two benchmark plots). The improvement is several orders of magnitude, demonstrating that downstream tasks benefit dramatically from our approach.**
>
> *One weakness is that the device was only simulated and not actually realized, so uncertainty remains with respect to practical performance (although the simulations are extensive, so I do not think this is a reason to reject). I think the paper would benefit from a Limitations section, which would make it clear what effects were not captured in the simulations. For example, what about PVT variations?*
>
> We agree that simulation-based results inherently have uncertainty and we will add more details in the Appendix for the camera-ready version on that. It is important to note that the prototypical manufacturing of these devices takes several years and requires significant material resources, especially in an academic setting. Considering these factors, we believe that sharing and comparing our results is both justifiable and beneficial for the broader community, particularly given the notable differences we have observed.
>
> We hope that we have successfully addressed your concerns!
>
> Kind regards,
>
> The Authors

---

> ### Comment · Reviewer_LDN7 · 2024-11-22
> **Reply to authors**
>
> Dear Authors,
>
> Thank you for the benchmark on rejection sampling, it addresses my concern as it is on a rather general task and the results are convincing, namely that the energy-efficiency advantage is largely conserved. I believe this addition improves the paper and have raised my score accordingly.

---

### Official Review · Reviewer_mR9r · 2024-10-30

**Soundness:** 2
**Presentation:** 1
**Contribution:** 1
**Rating:** 3
**Confidence:** 4

**Summary:**

This paper presents a framework to generate uniform Floating-point numbers with stochastically switching magnetic tunnel junction (MTJ) devices. A collection of devices produces correctly sampled mantissa and exponent bits by tuning their Bernoulli distribution according to a closed-form solution. The authors compare their MTJ approach with the pseudo-random number algorithms Mersenne-Twister/PCG. Through SPICE simulation and measurements (Antunes & Hil 2024, Noureddine, 2022), an energy reduction of 9721x/5649x is shown. In addition, the authors explain the construction of general distribution from uniform ones and methods to sum and multiply them. They also present the analysis of the approximation errors.

**Strengths:**

A potential low-energy integrated framework for large-scale random number generation is shown. The methodology is straightforward and applicable to any source of tunable Bernoulli distributions. One of the applications includes probabilistic machine learning.

**Weaknesses:**

The paper presents a general framework for random number generation with MTJ or any Bernoulli source. Random number generation is part of many machine learning methods, but it is not clear how this is specifically relevant to an ML audience.

The paper presents the energy consumption resulting from the SPICE simulation of the devices and parts of the additional circuitry needed. Whereas for the reference measurements for the pseudo algorithms, there’s no indication of what kind of intermediate computations and memory operations are being done. In addition, the comparison is between the generation of Float16 and Int32 where it is not elaborated on how this affects energy consumption. Overall it is not clear how it is a fair comparison.
The methodology is only compared against pseudo-random generation algorithms, but not against other hardware solutions. Other previous work on generating random numbers with MTJs (Example of R. Zhang et al. https://doi.org/10.1002/advs.202402182) are not discussed. Thus, there is not sufficient coverage of existing work.

**Questions:**

- How does it compare to other relevant works on hardware random number generation?
- What parts of the framework are novel and not seen in other works?
- How much would the energy consumption increase if the devices were integrated into modern architecture? In other words, what other consumption of energy is required?
- How much is the energy consumption reduction for relevant ML applications?
- Would the integrated MTJ devices have a similar lifetime to current hardware?

---

> ### Author Response · Authors · 2024-11-20
> **Updated Related Work Section & Clarification**
>
> Dear Reviewer mR9r,
>
> thank you for your feedback and taking the time to review our paper.
>
> We disagree with your overall perspective on the listed weaknesses. However, we acknowledge that the related work section needed the inclusion of additional s-MTJ-related works. We have updated the rebuttal version of the PDF accordingly and addressed this concern. We think that this revision might also contributes to your overall view of our paper.
>
> *Random number generation is part of many machine learning methods, but it is not clear how this is specifically relevant to an ML audience.*
>
> Machine learning algorithms have specific requirements for random number generation, making interdisciplinary approaches that combine ML and novel hardware essential:
> - First, they utilize the floating-point format (which differs from existing work on s-MTJs and many RNGs).
> - Second, while many machine learning algorithms currently rely on parametric distributions due to their low computational cost, this approach limits the expressiveness. Enabling arbitrary sampling is a significant progress for the machine learning audience.
> - Third, having a method for directly sampling from prior-likelihood or convolution distributions is highly relevant to a ML audience.
>
> *In addition, the comparison is between the generation of Float16 and Int32 where it is not elaborated on how this affects energy consumption. Overall it is not clear how it is a fair comparison.*
>
> Our research presents a novel sampling approach tailored to machine learning. Therefore, we compare it against the random number generators currently used in relevant existing frameworks.
>
> *Other previous work on generating random numbers with MTJs (Example of R. Zhang et al. https://doi.org/10.1002/advs.202402182) are not discussed. Thus, there is not sufficient coverage of existing work.*
>
> We agree that the previous coverage was insufficient. **We have updated the paper in the rebuttal version (see lines 109-124). Please refer to the updated PDF.**
> This version includes additional s-MTJ approaches and highlights differences to our work, including the example you provided. While the complete related work section is accessible in the PDF, I would like to elaborate on the differences between our work and that of Zhang et al.
> - They only consider random variables over integers, which makes them unsuitable for machine learning applications. We are proposing a floating-point approach.
> - Their approach samples sequentially for each bit, which results in a significant slowdown. In contrast, our method is constant and can be parallelized.
> - They do not take into account sampling from a product of distributions. We propose utilizing prior-likelihood and convolution transformations, allowing for direct sampling.
> - They continuously adjust write voltages for each bit and every sample. In our approach, we never change the write voltage, making it significantly more energy-efficient.
>
> *What parts of the framework are novel and not seen in other works?*
>
> All components of our framework presented in this paper are novel and have not been seen in previous works. While s-MTJs have been used before to generate random bits or distributions of integers, our approach—which includes direct energy-efficient floating-point generation, arbitrary floating-point sampling, and distributional transformations—has not been proposed before.
>
> Kind regards,
>
> The authors

---

> > ### Author Response · Authors · 2024-11-22
> > **ML Downstream Energy Reduction for ML Applications**
> >
> > Dear Reviewer mR9r,
> >
> > Please note that we have added a benchmark against rejection sampling to our paper. We have updated the PDF version of the paper, and you can find the benchmark results described in lines 400-431, which are supported by Appendix Section E (including pseudocode and two benchmark plots). The improvement is several orders of magnitude, demonstrating that downstream ML tasks/applications benefit dramatically from our approach.
> >
> > Kind regards,
> >
> > The Authors

---

> > > ### Comment · Reviewer_mR9r · 2024-11-25
> > >
> > > I appreciate the authors’ response. A discussion about previous work on RNG with MTJs has been newly added, and I see that the main contribution of the paper is the adoption of existing components to enable the sampling of uniform floats and other distributions.
> > >
> > > However, hardware RNG has been demonstrated with many different device concepts and is an application for unreliable memory devices. This has been an active area of research for many years, and in my opinion, the pieces introduced in this work are only an incremental addition to the previous work.
> > >
> > > I have also reviewed the supplementary materials and the citations, and concluded that the energy consumption comparison is unfair. The SPICE simulation compared to the energy consumption of a process on an AMD CPU is not sufficient.
> > >
> > > The authors did not address my other two questions:
> > > - How much would the energy consumption increase if the devices were integrated into modern architecture? In other words, what other consumption of energy is required?
> > > - Would the integrated MTJ devices have a similar lifetime to current hardware?

---

> ### Author Response · Authors · 2024-12-03
>
> *However, hardware RNG has been demonstrated with many different device concepts [...] This has been an active area of research for many years, [...] pieces introduced in this work are only an incremental addition to the previous work.*
>
> We do not claim to be the only or first ones working on hardware random number generators. However, our novel approach offers significant improvements for sampling-based machine learning applications and introduces novel techniques from both algorithmic and hardware perspectives. We see no evidence presented by you to suggest that we are merely making incremental improvements based on other work, including the mentioned paper by Zhang et al. We are furthermore utilizing new 2D materials that have only recently been observed to function electrically at room temperature and present a lightweight current steering digital-to-analog converter (DAC) that is specifically designed for our application.
>
> *I [...] concluded that the energy consumption comparison is unfair. The SPICE simulation compared to the energy consumption of a process on an AMD CPU is not sufficient.*
>
> We regret that you did not provide any actionable feedback, as no specific reasons were given to explain why you believe our benchmark is insufficient or under what conditions you think it would be considered sufficient. Considering the orders of magnitude in difference, we do not see that further benchmarks add fundamentally different information that changes the message of our paper.
>
> *How much would the energy consumption increase if the devices were integrated into modern architecture? In other words, what other consumption of energy is required?*
>
> While we acknowledge that an actual implementation will obviously not give the exact numbers we have calculated, the rigor of presented calculations and the orders of magnitude benefit observed make it clear that the proposed approach will result in clear and significant benefits.
>
> *Would the integrated MTJ devices have a similar lifetime to current hardware?*
>
> Endurance is in fact a key advantage of MTJs as compared to other forms of stochastic devices. MTJs are often touted to be having unlimited endurance. The marginal issue that arises in spin transfer torque based MTJs due to flow of switching current through the tunnel barrier is also milder in our proposed devices as the current needed for tilting stochasticity is much lower than the currents needed for error-free deterministic switching.
>
> Kind regards,
>
> The Authors

---

### Official Review · Reviewer_QDDw · 2024-11-04

**Soundness:** 3
**Presentation:** 2
**Contribution:** 2
**Rating:** 5
**Confidence:** 2

**Summary:**

This paper presents an energy-efficient algorithm for random sampling using stochastically switching magnetic tunnel junction (s-MTJ) devices. By directly mapping the physical properties of the s-MTJ to a uniform Float16 distribution, the proposed method avoids the computational overhead of symbolic calculations. Additionally, it introduces a method to sample from arbitrary 1D distributions using a mixture model approach, achieving low approximation errors.



Question:

**Strengths:**

1. The proposed framework is innovative, demonstrating both originality and significant potential in energy-efficient random sampling. The authors support these claims through simulations that indicate notable energy savings compared to existing methods.

2. By aligning the random generation process with the statistical properties of the Float16 format, this method sidesteps complex symbolic computations, enhancing both efficiency and simplicity.

3. By decomposing complex distributions into mixtures of uniform distributions, this approach allows for sampling from arbitrary 1D distributions without closed-form solutions, thus expanding the practical utility

**Weaknesses:**

1. The reliance on specific s-MTJ hardware may limit the method’s accessibility and applicability, particularly for researchers or practitioners who lack access to such specialized components, potentially requiring additional investment.

2. Due to physical constraints in setting bias currents and control bits, there may be small approximation errors in generating the intended Bernoulli distributions, which could impact applications requiring precise random number distributions.  It was also unclear if the ambient temperature of the chip would also impact the distributions (which is hard to control).

**Questions:**

How does the presence of genuine randomness affect model stability and repeatability in long-running applications, especially in training deep models with dropout or other probabilistic methods?

Can you further clarify how the s-MJT would implement equations the operations on the sampled distributions (Eqns 11-16). Is the idea that these equations are done in some way off-line such that these operations amount to sampling?  I think this can be more clear. It would also be useful to understand if any other operation is typically necessary, such as a non-linear operation.

---

> ### Author Response · Authors · 2024-11-14
> **Answer**
>
> Dear Reviewer QDDw,
>
> Thank you for reviewing our paper. We appreciate your time and would like to address your questions and discuss the noted weaknesses.
>
> *"The reliance on specific s-MTJ hardware may limit the method’s accessibility and applicability, particularly for researchers or practitioners who lack access to such specialized components, potentially requiring additional investment."*
>
> We believe it's important to distinguish between the current prototypical stage of our fundamental research setting and the subsequent adoption stage that will follow the publication of this approach. In the current stage, novel materials and devices are typically limited to those with specialized knowledge and resources; this stage is not intended for practitioners.
> In the adoption stage, companies such as NVIDIA, AMD, or Intel will integrate these components into their products (e.g., as a separate unit within a GPU) and make them available to researchers and practitioners. **This principle applies to any hardware-based device; we do not consider this a weakness of our paper**.
> To emphasize that this is a realistic pathway, we can point to deterministic MTJs, which, with similar materials and device structures, have already been adopted into commercial computer chips in the form of MRAMs. These components are now accessible to all researchers and practitioners.
>
> *"It was also unclear if the ambient temperature of the chip would also impact the distributions (which is hard to control)."*
>
> In general, s-MTJs are susceptible to process reliability and environmental effects just as any other semiconducting devices. Variation of temperature can alter natural frequency of the stochastic devices, as shown in Figure 4b of the manuscript. However, note again that deterministic MTJs have been successfully adopted into commercial computer chips and this is a solvable manufacturing-related problem.
>
> *"How does the presence of genuine randomness affect model stability and repeatability in long-running applications, especially in training deep models with dropout or other probabilistic methods?"*
>
> The repeatability of experiments is a legitimate concern when it comes to using genuine randomness. However, we believe that, in many real-world applications, cost-effectiveness is more important than repeatability. For instance, a search engine or a large online retailer that requires billions of inferences may prioritize efficiency over consistent results. It really depends on the individual use case.
> Also, the trade-off between repeatability and energy efficiency may need to be reconsidered in light of the significant energy consumption associated with current large-scale AI systems. However, addressing all those (legitimate) trade-offs falls outside the reasonable scope of our paper. Our focus is on proposing a new energy-efficient computing paradigm.
>
> *"Can you further clarify how the s-MJT would implement equations the operations on the sampled distributions (Eqns 11-16). Is the idea that these equations are done in some way off-line such that these operations amount to sampling? I think this can be more clear. It would also be useful to understand if any other operation is typically necessary, such as a non-linear operation."*
>
> Our paper introduces two novel sampling paradigms. First, we present highly energy-efficient uniform sampling. Second, we propose a mixture of uniforms to create a universal sampler capable of handling arbitrary distributions. Additionally, Equations 11-16 enable the application of convolutions and prior-likelihood transformations directly within the representation space of the distributions. These transformations can be implemented in two different ways, depending on how a hardware manufacturer chooses to apply our approach.
> The first option involves directly implementing the transformations as digital logic at the circuit level, near the s-MTJ devices. This approach allows for more efficient execution and integration into applications. The second way, if a direct implementation is not intended, is computing them at the software level while leveraging the highly efficient uniform sampling provided by the s-MTJ devices at the hardware level. In both scenarios, the transformations can be interpreted as a learning stage (or inference stage in probabilistic terminology) that occurs before sampling, as in other machine learning applications. We are happy to clarify that in the camera-ready version.
>
> We hope that we answered your questions and addressed your concerns! We would be happy if you give us your acceptance to present this new paradigm to our community!

---

> > ### Comment · Reviewer_QDDw · 2024-11-21
> > **Temperature dependence**
> >
> > In standard CMOS temperature dependence is handled using built-in margins associated with the design methodology. A clock frequency of a synchronous digital chip is set to accommodate range of delays of the devices over the intended temperature range of the chip. If the chip get's too hot (and delays get too long) then the clock frequency can be dynamically adjusted (or the system is designed to shut down - as my iPhone often does). It is unclear how this type of margin could be employed in your proposed paradigm in which you are relying on precise distributions that are effected by temperature. Can you elaborate on how this is a "solvable manufacturing-related problem"?
> >
> > Thank you for explaining that the potential use of digital logic in your paradigm. What is unclear to me is how much the digital logic may end up forming a lower bound on the energy savings improvement your paradigm provides. Can you somehow argue that the energy consumption of digital logic is somehow orders of magnitude lower than in standard ML inference engines?  I'm worried about Amdhal's law here and that while the sampling is efficient there are other aspects of the system that end-up limiting the overall efficiency of the system.

---

> ### Author Response · Authors · 2024-11-21
> **Temperature dependence and speed**
>
> *In standard CMOS temperature dependence is handled using built-in margins associated with the design methodology. A clock frequency of a synchronous digital chip is set to accommodate range of delays of the devices over the intended temperature range of the chip. If the chip get's too hot (and delays get too long) then the clock frequency can be dynamically adjusted (or the system is designed to shut down - as my iPhone often does). It is unclear how this type of margin could be employed in your proposed paradigm in which you are relying on precise distributions that are effected by temperature. Can you elaborate on how this is a "solvable manufacturing-related problem"?*
>
> We appreciate the reviewer’s thoughtful query regarding temperature variation and its potential impact on our proposed paradigm. In data center environments, thermal stability is actively managed to avoid dramatic temperature fluctuations, resulting in relatively small and predictable variations. These variations primarily affect the natural frequency and resistance of stochastic magnetic tunnel junctions (sMTJs). Importantly, minor resistance changes do not impact their operation as digital bits. As for the natural frequency, even if it increases from, for example, 1 GHz to 1.5 GHz across the operational temperature range, this frequency remains significantly higher than the typical sampling frequency (e.g., 100 MHz). As such, the sMTJ outputs remain uncorrelated, preserving the required randomness. Moreover, since devices are more likely to heat up in practice, the resulting increase in natural frequency would only improve the randomness, further strengthening the system’s performance under real-world conditions.
>
> *What is unclear to me is how much the digital logic may end up forming a lower bound on the energy savings improvement your paradigm provides. […]  I'm worried about Amdhal's law here and that while the sampling is efficient there are other aspects of the system that end-up limiting the overall efficiency of the system.*
>
> We believe the answer is twofold. First, in applications such as initializing weights in deep learning models or implementing dropout, our approach contributes to energy savings. This is particularly relevant given the global scaling of energy systems for AI, both in terms of costs and potential greenhouse gas emissions. While we achieve energy savings in these cases, there is likely no significant overall speedup.
> Conversely, algorithms that can be expressed through sampling might benefit significantly in terms of speedup. We anticipate that the probabilistic machine learning domain—for example, areas like Bayesian Neural Networks, which currently require massive resources and lack scalability—could see advantages. It's also worth noting that our approach is embarrassingly parallel in both sampling and prior-likelihood transformation. **However, the primary goal of this paper is energy-efficient sampling and not speed enhancement.**
>
> Kind regards,
>
> The Authors

---

> > ### Comment · Reviewer_QDDw · 2024-11-27
> > **Energy-efficiency analysis**
> >
> > I should clarify that my reference to Amdhal's law was not about speed-up but about the fact that more generally if you improve something that is say 20% of the whole you can only save 20%. My apologies for the confusion.
> >
> > So, my biggest concern is that random sampling may represent only a small fraction of the energy consumption of the real system. That is why I asked about the digital logic associated with equations [11-16]. My concern is that the energy consumption of the digital logic would limit the overall advantages that using sMTJ's would allow. I apologize for not being more clear, but can you clarify this point?
> >
> > I should also point out that this concern about overall energy savings seems to be shared by reviewer mR9r and that you have addressed this comment well for the rejection sampling problem described in lines 400-432 and in Appendix F. I do wish to ask one more thing, however. Figures 8 and 9 of Appendix F refer to "Other operations" and I could not find a clear description of what these entail. Is it related to the phrase "and according linear transformation"? Can you clarify what these operations are and what they are used for?

---

> > > ### Author Response · Authors · 2024-11-28
> > >
> > > Dear Reviewer QDDw,
> > >
> > > thank you for your comment and clarification! We are glad that our rejection sampling benchmark covered your concern.
> > >
> > > *I do wish to ask one more thing, however. Figures 8 and 9 of Appendix F refer to "Other operations" and I could not find a clear description of what these entail. Is it related to the phrase "and according linear transformation"? Can you clarify what these operations are and what they are used for?*
> > >
> > > Your understanding is indeed correct! The term "other operations" refers to the normalization described in Equation (9), which involves an additional five floating-point operations. By applying this normalization, we ensure that we can sample from any uniform range, such as the [0, 1] interval, rather than being limited to the full Float16 range, which is essential for real-world applications. This normalization overhead is included in all our benchmarks, and we have now clarified this in the corresponding figure captions.
> > >
> > > Kind Regards,
> > >
> > > The Authors

---

> > > > ### Comment · Reviewer_QDDw · 2024-12-02
> > > > **Final thoughts**
> > > >
> > > > I thank the author for addressing my remaining comments. However, as I am not an expert in MTJs (but have a general hardware and ML background), I would like to hear the author's response to reviewer's mR9r last comment about the fairness of the energy comparison as well as comment about alternative hardware-based RNGs before adjusting my score.

---

> ### Author Response · Authors · 2024-12-03
>
> Dear Reviewer QDDw,
>
> Thank you for your time and thorough evaluation of our work. We would like to clarify our position regarding the comments from reviewer mR9r. We emphasize that we see no evidence suggesting that additional benchmarks would provide any new information that alters the fundamental message of our paper, given the significant differences in magnitude. We also believe that we have adequately addressed the reviewer’s concerns regarding alternative hardware-based RNGs in our previous responses and have updated the related work section in the paper to highlight the similarities and differences between our work compared to both s-MTJ and other RNG approaches. In general, the last response (https://openreview.net/forum?id=HkPz96fgOv&noteId=4qmMwZi605) did not offer any further actionable feedback for us by the reviewer aside from „concluding“ remarks. We disagree with those opinions and believe the statements lack solid backing.
>
> Kind regards,
>
> The Authors

---

### Meta-Review · Area_Chair_Kspv · 2024-12-19

**Metareview:**

The authors of this paper propose a simulation method for generating random numbers more efficiently, which could significantly benefit most Bayesian algorithms—particularly those that do not rely on approximations—where sampling often becomes a bottleneck in handling larger datasets or achieving more accurate posterior estimates. The reviewers, however, were divided and could not reach a consensus. Reviewer mR9r raised compelling concerns about the limitations of this work, which were not adequately addressed in the authors' responses. Reviewer QDDw supported mR9r's critique. The paper must thoroughly address these issues before it can be considered for publication at ICLR or any other top-tier machine learning conference.

During my discussions with the reviewers, I came across a 2020 article published in Physical Review Letters (PRL) with a very similar title, which is not referenced in this paper. Although the two papers address different aspects, the overlap in topics and titles warrants acknowledgment. This paper should include a discussion highlighting the differences between the two works to provide clarity and properly situate its contributions in the broader context of the field https://journals.aps.org/prapplied/abstract/10.1103/PhysRevApplied.13.034016.

Finally, the authors should consider that while random number generators play a critical role in machine learning research, the field itself does not typically focus on developing new hardware implementations. If the proposed improvement is not made readily accessible for practical use, its impact on machine learning research would be minimal. Ensuring usability and accessibility should therefore be a priority to maximize the paper's relevance and contribution to the field.

**Additional Comments On Reviewer Discussion:**

The authors responded and the reviewers engage with the authors, but only the negative reviewers reacted to my questions. The positive reviewers were more shallow and they did not champion the paper.

I also have concerns about ICLR being the right venue for this paper, as it does not give us anything to work with.

---

### Decision · Program_Chairs · 2025-01-22

Reject